# Influence of Organic Loading Rate on Methane Production from Brewery Wastewater in Bioelectrochemical Anaerobic Digestion

Hongda Pan, Qing Feng *, Yong Zhao, Xiaoxiang Li and Hao Zi

College of Environmental Science and Engineering, Qilu University of Technology (Shandong Academy of Sciences), Jinan 250353, China; 18365752770@163.com (H.P.); 19546158258@163.com (Y.Z.); 15863195913@163.com (X.L.); zihao9069@163.com (H.Z.)

* Correspondence: qingfeng@qlu.edu.cn

**Abstract:** The effect of bioelectrochemical anaerobic digestion (BEAD) on the methanogenic performance of brewery wastewater at different organic loading rates (OLRs) was investigated and compared to conventional anaerobic digestion. A continuous BEAD reactor was used to treat brewery wastewater at different OLRs of 2, 4, 8, 16, and 20 g COD/L.d. The experimental results showed that the methane production was gradually increased from 0.48 L/L.d at an OLR of 2 g COD/L.d to 5.64 L/L.d at an OLR of 20 g COD/L.d. The methane production of the BEAD system was significantly higher than that of the conventional anaerobic reactor, indicating that BEAD has a better treatment effect for brewery wastewater. The performance of the conventional anaerobic reactor was significantly reduced especially at an OLR of 16 g COD/L.d, while the BEAD system could withstand a higher OLR. Bioelectrochemical systems provide a completely new platform for the anaerobic treatment of brewery wastewater and greatly improve the operation of anaerobic processes.

**Keywords:** bioelectrochemical system; anaerobic reactor; OLR; brewery wastewater; methanogenic performance

## 1. Introduction

China is a large brewery-producing and -consuming country, with the brewing industry being the fifth largest sector in the beverage industry in terms of annual sales [1]. According to the Ministry of Industry and Information Technology, approximately 33.13 million m$^3$ of brewery was produced in China between January and October 2021, an increase of 5% in production compared to the same period last year. It is estimated that the production of each litre of brewery generates 3–10 L of brewery wastewater [2]. Brewery wastewater mainly comes from the brewery production process of soaking wheat, saccharification, fermentation and filtration, filling, and other processes, and it mainly contains sugars, alcohols, yeast residues, proteins, and volatile fatty acids (VFA), etc. [3]. The aim of brewery wastewater treatment methods is to remove harmful compounds from the wastewater so that it can be safely discharged into the environment. This can be achieved through physical, chemical, and biological treatment of brewery wastewater, and in most cases, these treatment methods are used in combination with each other. Physical treatment methods include membrane filtration processes (nanofiltration, reverse osmosis, electrodialysis) and adsorption techniques. Chemical treatment methods may involve pH adjustment, coagulation, or flocculation. Biological treatment methods may involve aerobic treatment and anaerobic treatment [4].

Breweries produce large amounts of wastewater containing high concentrations of degradable organic pollutants, which are great fermentation substrates for $CH_4$ and $H^2$. Bioelectrochemical systems (BESs) are considered unique, sustainable, and environmentally friendly processes. Chemical energy obtained by BESs from wastewater and lignocellulosic biomass is converted to electrical energy, hydrogen, and value-added bioelectrochemistry

through a redox reaction process using bacteria as biocatalysts [5]. BEAD, a system that combines BESs with anaerobic digestion (AD), has been proposed and shown to hold promise for improving the treatment of organic matter in biodegradation and biogas production [6,7]. Higher process efficiency, shorter stabilisation periods, and faster methane production can be achieved using electrochemically activated microorganisms [8–10]. BEAD can achieve higher process efficiencies, shorter stabilisation periods, and increased methane yields using electrochemically activated microorganisms compared to conventional AD [11]. Extensive engineering optimisation studies have also been carried out to improve the scalability and economics of BEAD, including applied voltage, electrode materials (carbon-based and stainless steel materials), electrode spacing, etc. [12,13]. Also, the organic loading rate (OLR) is a key operational parameter affecting BEAD performance [14].

OLR is a crucial parameter to consider when operating a BEAD system [14]. It represents the amount of organic matter added to the system. A higher OLR, combined with a shorter hydraulic retention time (HRT), is often preferred because it leads to a smaller reactor size, lower energy consumption for heating, and improved methane production efficiency. However, if the OLR exceeds a certain threshold, it can cause system instability [15]. Numerous studies have investigated the impact of increasing OLR on AD and biogas production. It was observed that biogas production tends to improve within a certain range of OLR values [16]. Controlling the OLR can be a useful engineering strategy to optimise methane production and ensure system stability [17,18]. However, there has been a limited focus on the specific effects of OLRs on anaerobic digestion and methane production. The optimal OLR for methane production may vary depending on the type of organic matter being treated [18]. For example, one study showed that 100 (mL $CH_4$/g chemical oxygen demand (COD)) could be achieved at an OLR of 3 g COD/L.d, and further increased to 400 (mL $CH_4$/g COD) at an OLR of 12 g COD/(L.d). However, further increasing the OLR to 15 g COD/L.d resulted in a reduction in methane production [19]. Another study observed an increase in OLR from 5 (g/L.d) total solids (TS) to 11 (g/L.d) TS and reported changes in product composition [19,20]. The OLR also plays a role in bioelectrochemical anaerobic digestion and methane production from brewery wastewater [21]. Research has shown that monitoring the OLR and pH can facilitate the selective production of organic acids during the anaerobic acidogenesis process [22]. An increase in OLR has been reported to have an impact on microbial community structure and metabolic pathways, leading to different distributions of metabolic products [23]. The delivery of VFAs is an important consideration when utilising VFA-rich fermentation liquid for various applications, such as wastewater treatment [23]. Additionally, specific VFAs have been found to promote the synthesis of certain polyhydroxyalkanoates (PHA). For example, acids with an even number of carbon atoms favour the synthesis of 3-hydroxybutyrate and promote the production of 3-hydroxyvalerate [19]. Therefore, to achieve high methane production and ensure system stability, it is essential to optimise the OLR based on the characteristics of the substrate being treated [24]. It is worth noting that no specific research on high OLR in BEAD systems for brewery wastewater has been reported thus far. This study provides more insight into bioelectrochemical anaerobic digestion and is expected to advance the development of bioelectrochemical anaerobic digestion technology.

In this study, bioelectrochemical enhanced methane production from brewery wastewater at different OLRs was investigated. Methane yield, electrochemical characteristics, state variables (pH, alkalinity, VFA), COD, and sulphate removal from anaerobic digestion were studied and compared with conventional anaerobic digestion. The objective of this study was to find out the optimum conditions for the biomethanation of brewery wastewater to make resourceful use of brewery wastewater possible.

## 2. Materials and Methods

### 2.1. Seed Sludge and Brewery Wastewater

The brewery wastewater was collected from the brewery at Qilu University of Technology (Changqing campus). The COD and pH of the brewery wastewater were 34.8 ± 1.6 g/L

and 3.6 ± 0.2, respectively. Anaerobic sludge was collected from an anaerobic digester for seed sludge (Jinan, China), and the COD and pH were 18.2 g/L and 6.93, respectively. Their characteristics are shown in Table 1.

**Table 1.** Characteristics of the anaerobic sludge and brewery wastewater.

| Parameters | Brewery Wastewater | Anaerobic Sludge |
|:---:|:---:|:---:|
| pH | 3.6 ± 0.2 | 6.93 |
| Alkalinity (mg/L as $CaCO_3$) | - | 1052 |
| COD (g/L) | 34.8 ± 1.6 | 18.2 |
| TS (g/L) | 28.3 ± 1.8 | 14.3 |
| VS (g/L) | 22.9 ± 1.6 | 7.6 |
| Sulphate (g/L) | 1.5 ± 0.5 | - |

### 2.2. Electrode Fabrication

The electrode materials of carbon cloth and carbon nanotube (CNT) were soaked in concentrated nitric acid (60% *v/v*) for 1 day and then rinsed with tap water to remove impurities and improve the hydrophilicity of the material surface. The carbon cloth was precipitated in a certain concentration of 1000 mg/L nickel dichloride solution and then electrophoretically treated at 30 V according to a previous study [25]. A uniform nickel layer was formed on the surface of the carbon cloth by electrophoretic deposition [26–28]. Then, 0.1 g of nickel dichloride, 1 g of carbon material, and Nafion binder were mixed and pasted onto the surface of the carbon cloth. The modified carbon cloth was pressed with a hot press at 200 °C and finally immersed in $C_{12}H_{25}SO_4Na$ (1%) solution to improve hydrophilicity and complete the carbon cloth electrode [25]. Separator and electrode assembly (SEA) were obtained by inserting a separator of a polypropylene nonwoven sheet between an anode and cathode of the same size. The reactor was supplied with 0.5 V from a DC power supply.

### 2.3. Reactor Design and Operation

The 5.5 L upflow anaerobic reactor (Figure 1) was configured from cylindrical acrylic resin. A funnel-shaped vessel was fixed at the bottom of the reactor and connected to an inlet valve to ensure even distribution of the wastewater. An outlet valve was installed at the top of the upflow anaerobic reactor to prevent leakage of biogas into the outlet valve by connecting a U-shaped pipe. An acrylic platform was installed on top of the upflow anaerobic reactor, which had three ports for biogas venting, liquid sampling, and biogas sampling. An acrylic tube submerged in a solution was connected to the bottom of the liquid sampling port in the cover. The biogas sampling port was sealed with an n-butyl rubber plug, while the biogas collection port was connected to an alternative water and gas collector via a rubber tube. To minimise the dissolution of biogas, the gas collector was filled with an acidic solution of saturated salt [25]. The electrode surface area was fixed at 20 m$^2$/m$^3$ and the temperature was kept at 35 ± 2 °C [29]. The influent COD was kept at 2 g/L, 4 g/L, 8 g/L, 16 g/L, and 20 g/L by diluting the brewery wastewater with different multiples. Anaerobic sludge was inoculated at 30% of the effective volume in the reactor. Brewery wastewater was continuously supplied at an OLR of 2 g COD/L.d, 4 g COD/L.d, 8 g COD/L.d, 16 g COD/L.d, and 20 g COD/L.d. Condition variables (pH, alkalinity, and VFA), biogas production, methane content, and organic removal rates were monitored to compare and evaluate the performance of the reactors.

### 2.4. Measurements and Calculations

During operation of the bioelectrochemical and UASB reactors, biogas was generated once per day, and biogas populations were analysed once per day using a GC (Series 580, Gow-Mac Instrument Co., Bethlehem, PA, USA) equipped with a thermal conductivity detector and a Porapak Q separator (6 ft × 1/8″ SS). The specific measurement conditions were as follows: the packed column model was TDX-01; the carrier gas was nitrogen; the

flow rate was 30 mL/min; the split ratio was 1:1; and the temperatures of the injector, the column box, and the detector were 150 °C, 120 °C, and 180 °C, respectively; and the sample was injected into the column at 1 mL each time. The amount of mesophilic acid was converted to standard temperature and pressure (STP) using the following Equation (1) [29].

$$V_{CH_4}(\text{at STP mL}) = V_{CH_4}(\text{at T, mL}) \times \frac{273}{273 + T} \times \frac{760 - W}{760}, \tag{1}$$

where T is the operating temperature of the anaerobic reaction set (0 °C) and W is the saturated water vapour pressure (mmHg) at the operating temperature. Using the measured current, the amount of methane produced by the reduction electrode of the BEAD reactor was estimated using the following Equation (2) [30].

$$P_c(\text{mL}) = \frac{\int_0^t i \, dt}{nF} \times VPM, \tag{2}$$

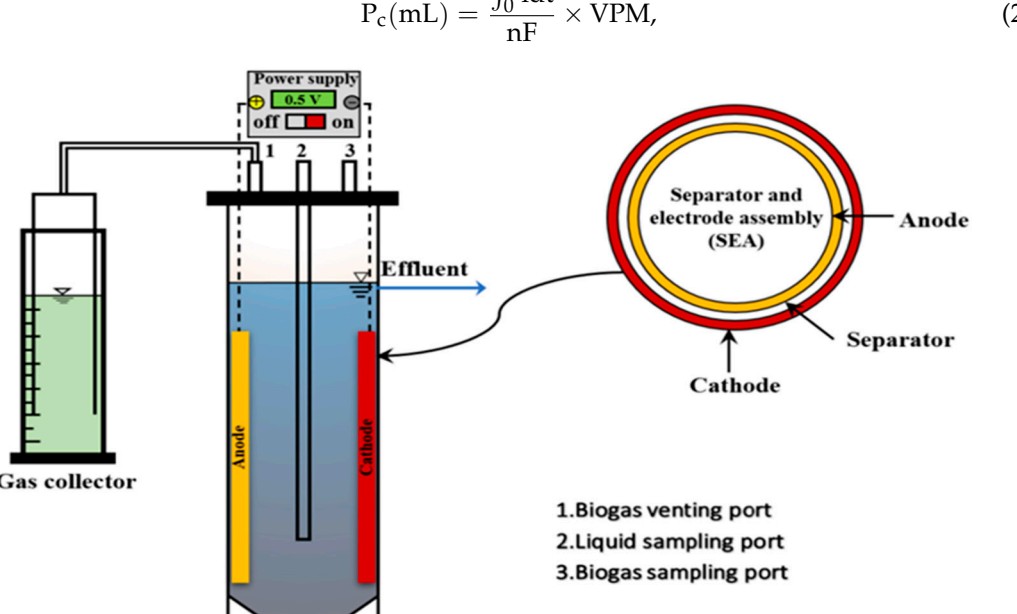

**Figure 1.** Reactor diagram.

A digital multimeter (model cDAQ-9174, National Instruments, Austin, TX, USA) was used to observe the current (i, A) through the external circuit of the BEAD reactor. The time at which the current was observed was noted as the variable t. In this experiment, the number of electrons per 1 mole of methane was considered to be n, and Faraday's constant F (96,485 C/mol) was used. The standard volume VPM represents the volume of 1 mole of methane and has a value of 22,400 mL/mol.

The pH and alkalinity of the wastewater were analysed using a pH meter (Orion Model 370) and standard methods (2005), measured once a day for the anaerobic reactor. A high-performance liquid chromatograph HPLC (DX-500, Varian, Palo Alto, CA, USA) with an Aminex HPX-87H separator was also used for analyses. Measurements of COD and sulphate were carried out according to standard methods (2005) and analysed twice a week (APHA, 2005). The current density, i.e., the observed current divided by the effective volume of the bioelectrochemical reactor, was calculated by observing the current in the external circuit of the bioelectrochemical reactor using a digital multimeter (cDAQ-9174, National Instruments, Austin, TX, USA). To assess the electrochemical performance of the anode and cathode, electrochemical impedance spectroscopy (EIS) and Tafel plots were also used. Electrochemical impedance spectroscopy experiments were carried out in the

open-circuit condition over a frequency range of 10 kHz to 10 MHz and using a Ag/AgCl electrode (RE-1B, ALS Co., Ltd., Tokyo, Japan) as a reference electrode.

Based on the results of the electrochemical impedance spectroscopy experiments, it was shown that the ohmic and capacitive impedances of the solution were connected in series. The electron transfer resistance and Faraday impedance consisted of the electron transfer resistance and were obtained in parallel with the Randle equivalent circuit model to obtain the Warburg element. By plotting the current–voltage experimental results as a Tafel curve, it is possible to determine the Tafel slope, i.e., the slope of a straight line with a coefficient of determination (r2) greater than or equal to 0.999.

## 3. Result and Discussion

### 3.1. State Variables (pH, Alkalinity, and VFA) under Different OLRs

The pH of the unneutralised brewery wastewater is as low as 3.6 to 4.0 and the wastewater is acidic. At an OLR of 2 g COD/L.d, the reactor stability decreases. To improve reactor stability, the effluent water containing alkalinity can be recycled to 1.5 times the influent water and used as an operating strategy for the BEAD reactor (Figure 2a). On day 23 of operation, the OLR was increased to 4 g COD/L.d. When the pH increased, the pH of the BEAD reactor effluent temporarily decreased to 6.98 but returned to a stable value of 7.3 or higher after 10 days (Table 2). When the OLR is increased to 8 g COD/L.d, the pH of the reactor effluent also drops temporarily, but returns to a stable value of 7.27 after about 10 days. In general, the optimum pH range for methane production in anaerobic digesters is 6.8~7.8, and it is recommended to maintain a pH of 7.0 or higher [31,32]. In this study, the pH changes of BEAD reactor were remained stable even when the OLR exceeded 16g COD/L, while the pH changes of control reactor were reduced rapidly (Figure 2a). Similar to pH, the alkalinity of the BEAD reactor effluent decreased slightly as the OLR increased (Figure 2b). At an OLR of 20 g COD/L.d, the alkalinity of the reactor remained stable at 5366 mg/L as the concentration of $CaCO_3$ increased to 2 g COD/L.d. The alkalinity of the effluent decreased slightly to 5241 mg/L (Table 2). The OLR was 20 g COD/L.d, and after 10 days, the alkalinity in the form of $CaCO_3$ was stable at 4867 mg/L. In general, the optimum alkalinity range for anaerobic digesters is 4000~7000 mg/L $CaCO_3$. The results imply that even at an OLR of 4 g COD/L.d, the alkalinity of the BEAD reactor remains stable at 4867 mg/L. The alkalinity of the BEAD reactor remained within the normal range.

VFA is an intermediate product of anaerobic digestion and is a methane precursor. Its nature and concentration are important indicators to evaluate the state of the anaerobic digester. At an OLR of 8 g COD/L.d, the VFA concentration in the BEAD reactor was 334 mg/L, which was comparable to the COD concentration and remained stable (Figure 3). However, at an OLR of 16 g COD/L.d, the VFA concentration increased approximately twofold to 703 mg/L.d and increased to 831 mg/L.d as the COD concentration increased. In general, the effect of VFA on alkalinity in a conventional anaerobic digester is 0.3 or less, and when the ratio of VFA to alkalinity exceeds 0.4, the anaerobic digester is considered to be in an unstable condition. At an OLR of 20 g COD/L.d, the VFA concentration was high but the VFA-to-alkalinity ratio was less than 0.17 (Table 2). This indicates that the BEAD reactor can operate stably at an OLR of 20 g COD/L.d and is not affected by the inhibition of VFA or process instability. As the OLR increased, the main components of VFA in the BEAD reactor were formic acid, acetic acid, propionic acid, and butyric acid, and their effect on the OLR was not significant. As the OLR and formic acid concentration increased, the NADH/NAD+ ratio and the concentration of acetic acid readily available to methanogenic bacteria in the BEAD reactor also increased. This may be due to the relatively lower pH of the BEAD reactor at high OLRs, resulting in a slight decrease in the acetic acid activity of the methanogenic bacteria.

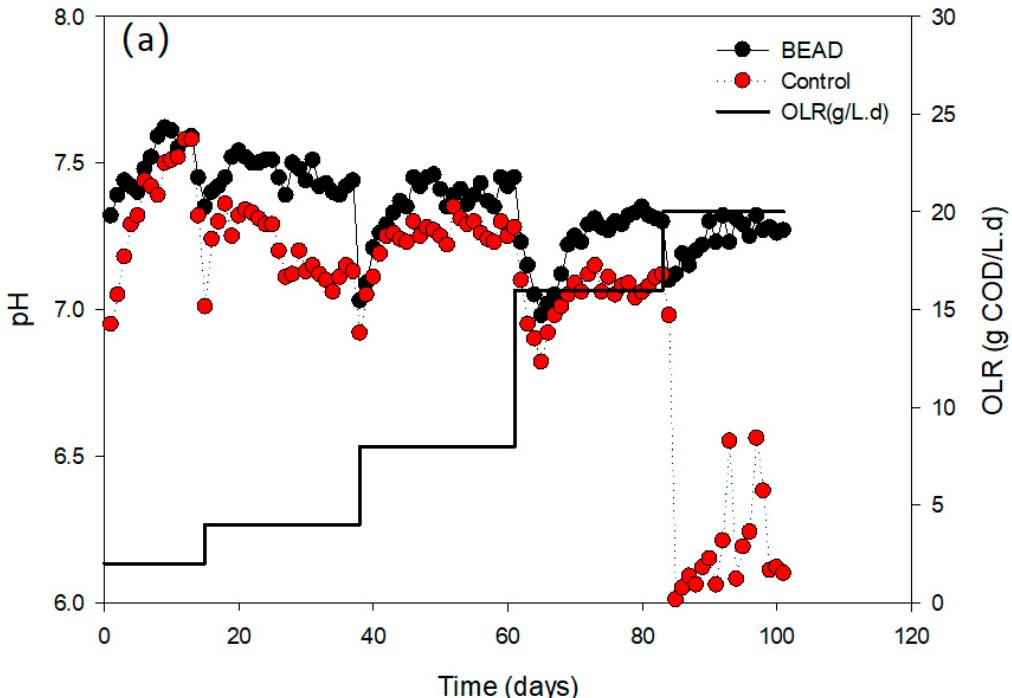

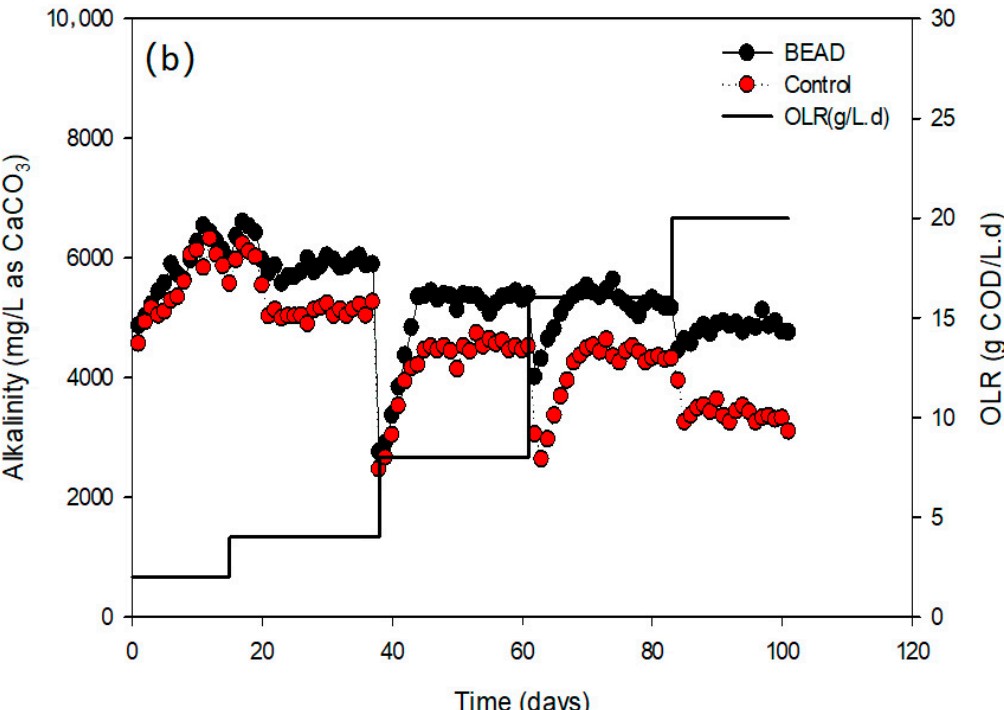

**Figure 2.** Changes in (**a**) pH and (**b**) alkalinity for BEAD reactor at different OLRs.

**Table 2.** Performance of BEAD reactor at different OLRs.

| OLR (COD/L.d) | 2 | | 4 | | 8 | | 16 | | 20 | |
|---|---|---|---|---|---|---|---|---|---|---|
| Reactors | BEAD | Control | BEAD | Control | BEAD | Control | BEAD | Control | BEAD | Control |
| pH | $7.58 \pm 0.03$ | $7.55 \pm 0.04$ | $7.41 \pm 0.02$ | $7.11 \pm 0.04$ | $7.42 \pm 0.05$ | $7.25 \pm 0.03$ | $7.34 \pm 0.01$ | $7.07 \pm 0.02$ | $7.27 \pm 0.04$ | $6.29 \pm 0.22$ |
| Alkalinity (mg/L as $CaCO_3$) | $6374 \pm 131$ | $6084 \pm 202$ | $5933 \pm 79$ | $5104 \pm 88$ | $5366 \pm 57$ | $4536 \pm 66$ | $5241 \pm 63$ | $4339 \pm 69$ | $4867 \pm 66$ | $3325 \pm 24$ |
| VFAs (mg/L as COD) | $864 \pm 102$ | $931 \pm 152$ | $719 \pm 52$ | $1627 \pm 135$ | $334 \pm 57$ | $3524 \pm 256$ | $703 \pm 72$ | $4357 \pm 539$ | $831 \pm 64$ | $5690 \pm 340$ |
| VFA/alkalinity | 0.14 | 0.15 | 0.12 | 0.32 | 0.06 | 0.78 | 0.13 | 1.00 | 0.17 | 1.71 |
| COD removal rate (%) | $94.0 \pm 0.4$ | $92.4 \pm 0.4$ | $96.0 \pm 0.4$ | $88.3 \pm 0.3$ | $96.5 \pm 0.1$ | $87.9 \pm 0.2$ | $91.3 \pm 0.2$ | $84.7 \pm 0.4$ | $86.3 \pm 0.4$ | $54.8 \pm 0.6$ |
| Sulphate removal rate (%) | $87.5 \pm 1.8$ | $59.1 \pm 3.1$ | $91.5 \pm 0.8$ | $55.7 \pm 5.1$ | $91.3 \pm 1.6$ | $64.1 \pm 5.2$ | $88.3 \pm 2.6$ | $57.2 \pm 2.0$ | $84.9 \pm 1.5$ | $48.2 \pm 5.7$ |
| Specific methane production rate (L/L.d) | $0.48 \pm 0.01$ | $0.43 \pm 0.01$ | $1.54 \pm 0.02$ | $1.00 \pm 0.06$ | $3.43 \pm 0.06$ | $2.18 \pm 0.02$ | $5.18 \pm 0.05$ | $3.53 \pm 0.05$ | $5.64 \pm 0.09$ | $2.04 \pm 0.01$ |
| Methane content (%) | $86.3 \pm 1.3$ | $77.1 \pm 0.4$ | $78.0 \pm 0.9$ | $56.0 \pm 1.8$ | $82.0 \pm 0.5$ | $68.2 \pm 0.8$ | $80.0 \pm 0.5$ | $64.8 \pm 0.5$ | $78.2 \pm 1.6$ | $55.9 \pm 1.8$ |
| Methane yield (mL $CH_4$/g COD) | $240 \pm 8$ | $217 \pm 4$ | $385 \pm 4$ | $250 \pm 15$ | $400 \pm 7$ | $272 \pm 3$ | $343 \pm 4$ | $221 \pm 3$ | $352 \pm 6$ | $127 \pm 10$ |

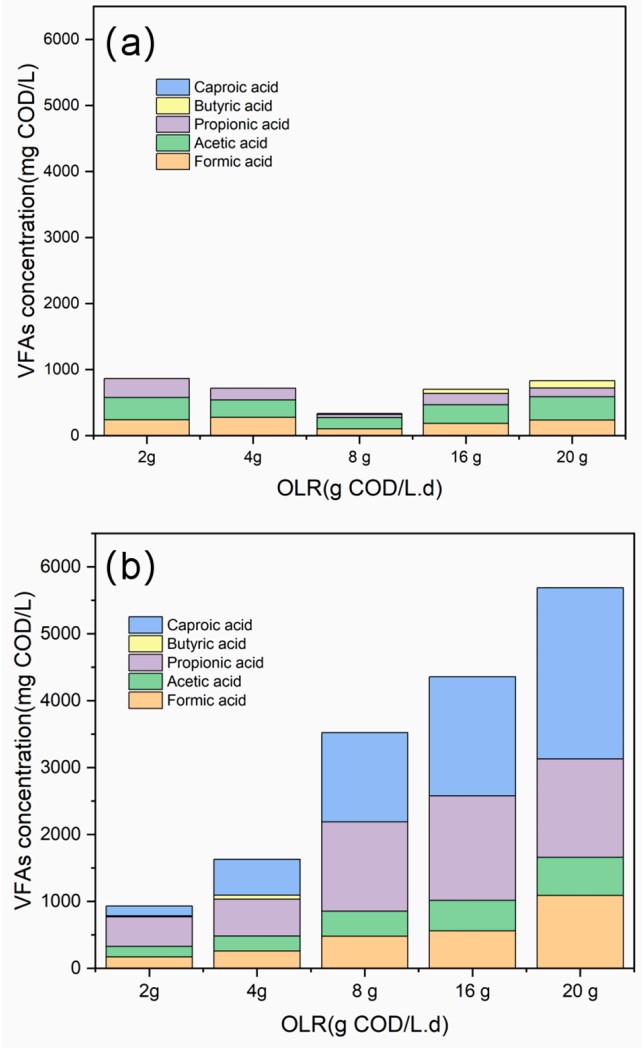

**Figure 3.** Levels of VFA components at different electrode areas in the (**a**) BEAD reactor and (**b**) control reactor.

### 3.2. COD and Sulphate Removal under Different OLRs

As shown in Figure 4a, the highest COD removal rate of 96.5% was achieved at an OLR of 2 g COD/L.d. When the OLR was increased to 4 g COD/L.d, the COD removal rate decreased to 91.3%. When the OLR was further increased to 8 g COD/L.d, the COD removal rate decreased to 86.3%. However, as shown in Table 2, even at high OLRs (16 g COD/L.d), the COD removal rate for the UASB reactor was approximately 6.9% at an organic loading rate of 20 g COD/L.d. This is due to the effect of the DIET reaction of the electroactive microorganisms activated in the BEAD reactor. Figure 4b shows that the highest sulphate removal rate of 91.3% was achieved at an OLR of 8 g COD/L.d. When the OLR was increased to 16 g COD/L.d, the sulphate removal rate was 88.3%. When the OLR was further increased to 20 g COD/L.d, the sulphate removal rate decreased to 84.9%. As the OLR increased, the COD removal rate gradually decreased. This may be due to competition between sulphate-reducing bacteria and methanogenic bacteria for substrate. However, the results showed that at higher OLRs, the activity of the sulphate-reducing bacteria also gradually decreased with decreasing pH and alkalinity.

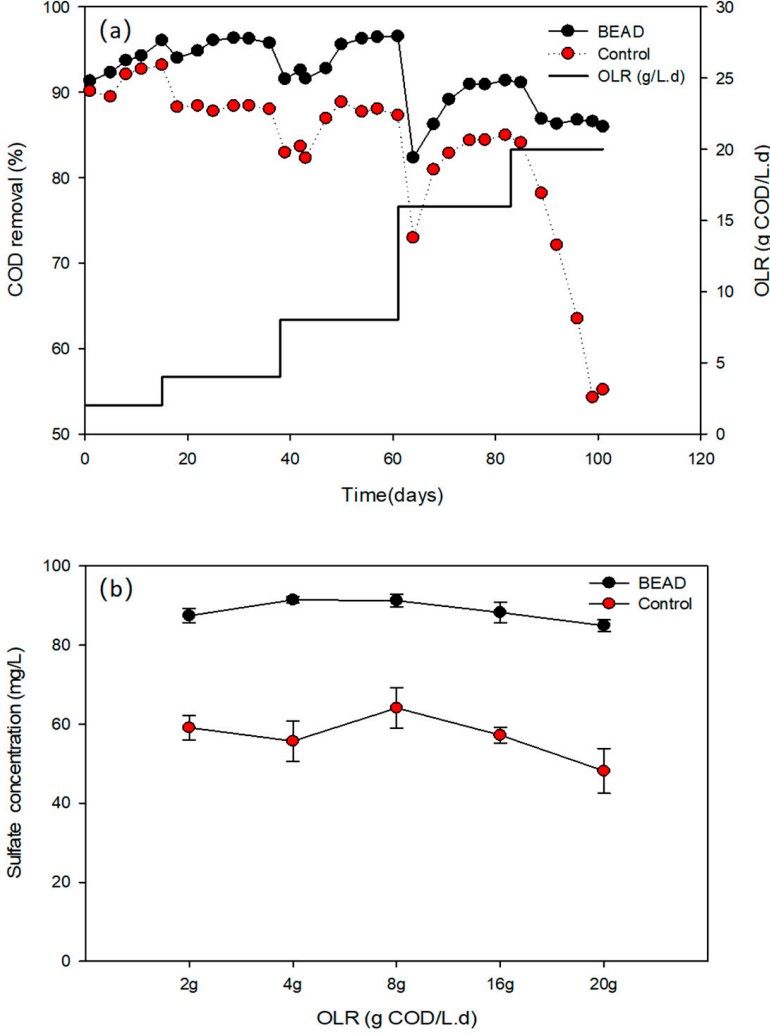

**Figure 4.** (**a**) COD removal efficiency and (**b**) sulphate removal efficiency in the BEAD reactor at different OLRs.

### 3.3. Methane Production under Different OLRs

As shown in Figure 5a, there was an increasing trend in methane production with increasing reactor operating time. At an OLR of 2 g COD/L.d, the specific methane yield was 3.82 L CH$_4$/L.d (Table 2). However, when the OLR was increased to 4 g COD/L.d,

methane production from the reactor increased proportionally with the increasing feed-water loading ratio and stabilised at 5.78 $CH_4$/L.d. Similarly, when the OLR was increased to 8 g COD/L.d, methane production was 6.35 $CH_4$/L.d, an increase of approximately 66.2% compared to 16 g COD/L.d.

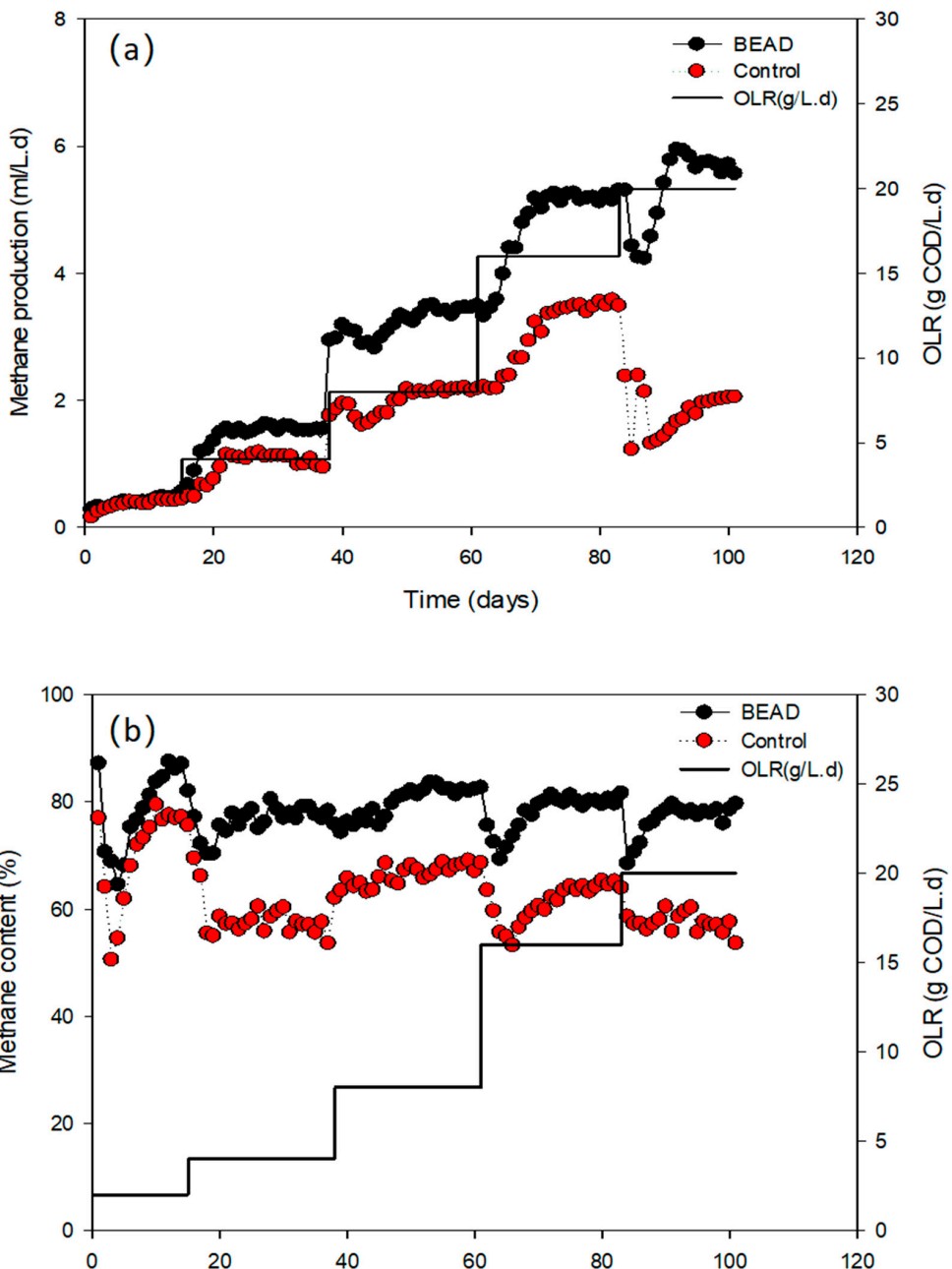

**Figure 5.** (**a**) Specific methane production rate and (**b**) methane composition in the BEAD reactor at different OLRs.

Figure 5b shows the biogas methane content plotted against the OLR. In the BEAD reactor, the methane content was stable at around 82.1% at an OLR of 20 g COD/L.d. However, at an OLR of 2 g COD/L.d, the methane content was about 80.3%, and at an OLR of 4 g COD/L.d, the methane content was about 78.1%.

In an anaerobic digester, the methane content depends on the substrate and the pH. pH affects the conversion efficiency of electrons to methane and carbon dioxide. As the OLR

increases, the methane content of the biogas may decrease, partly due to the decrease in pH with an increasing loading rate. In addition, it can be assumed that the proportion of the methanogenic process that is accounted for by the DIET reaction also gradually decreases.

The methane yield is the amount of methane produced per gram of COD removal in the wastewater. In this study, methane production was also measured at different OLRs. However, when the OLR was increased to 4 g COD/L.d, the methane yield decreased to 396 mL $CH_4$/g CODr. Further increasing the OLR to 8 g COD/L.d further reduced the methane yield to 365 mL $CH_4$/g CODr. This indicates that the methane yield decreases with increasing OLR and is consistent with the trend in methane content in biogas.

It was also found that electron transfer in inoculation for methanogenesis via intermediate products such as hydrogen and formic acid had greater electron transfer losses than direct electron transfer DIET [32–35]. Therefore, the indirect electron transfer methanogenesis rate gradually increased with increasing OLRs. It should be noted that the methane yield obtained in this study (365 mL $CH_4$/g CODr) was higher than the expected yield of the anaerobic reaction (350 mL $CH_4$/g CODr) calculated theoretically using the Buswell equation, which indicates the better performance of the reactor. In addition, energy efficiency is defined as the percentage of methane energy recovered to the wastewater substrate and electrical energy consumed in the reactor. The experimental results showed that the reactor had the highest energy efficiency of 97.7% at an OLR of 16 g COD/L.d. However, at an OLR of 20 g COD/L.d, the energy efficiency decreased slightly to 95.5%. At an OLR of 20 g COD/L.d, the energy efficiency of the reactor dropped to 88.9%. Therefore, considering both methane yield and energy efficiency, the reactor performs best at an OLR of 8 g COD/L.d. Finally, it is worth noting that the reactor was able to maintain stable operation and high methane yields even at an OLR of 20 g COD/L.d. This indicates that the reactor has good stability and high efficiency.

### 3.4. Electrochemical Characteristics of Electrodes under Different OLRs

Based on the results of the electrochemical impedance spectroscopy experiments (Figure 6) and the data in Table 3, some effects of the organic loading ratio on the anode and cathode can be observed. Firstly, the ohmic resistance of the anode fluctuated between 1.36 and 1.85 $\Omega$, with a nonsignificant difference as the organic loading ratio increased (Table 3). However, the electron transfer resistance (Rct, $\Omega$) of the anode was 5.88 $\Omega$ when the organic loading ratio was 2 g COD/L.d, but it increased to 7.43 $\Omega$ (Figure 6a) when the organic loading ratio was increased to 4 g COD/L.d. When the organic loading ratio reached 8 g COD/L.d, Rct reached a maximum value of 7.98 $\Omega$. Electron transfer resistance is a parameter related to the activation energy of the electrode reaction and plays a role in electrochemical reactions at the electrode surface. This suggests that the catalytic activity of electroactive microorganisms attached to and growing on the anode surface decreases at high OLRs, possibly due to the effects of factors such as high temperatures and incomplete neutralisation of the feed water acidity. To address this issue, the reactor can be operated by increasing the organic loading ratio and increasing the effluent circulation rate. On the other hand, the electron transfer resistance of the cathode varied between 1.65 and 2.35 $\Omega$, which was significantly lower than that of the anode and was not significantly affected by the increased OLR (Figure 6b). This indicates that the rate-limiting step of the electrode reaction occurs at the anode. As for the capacitive impedance and Warburg resistance of the cathode, the values are greater at the cathode than at the anode, but there is no significant trend at different OLRs. In summary, based on the experimental results, it can be seen that the OLR has a significant effect on the electron transfer resistance at the anode and cathode, whereas it has less effect on other parameters. These results are important for optimising the reactor performance and operating conditions.

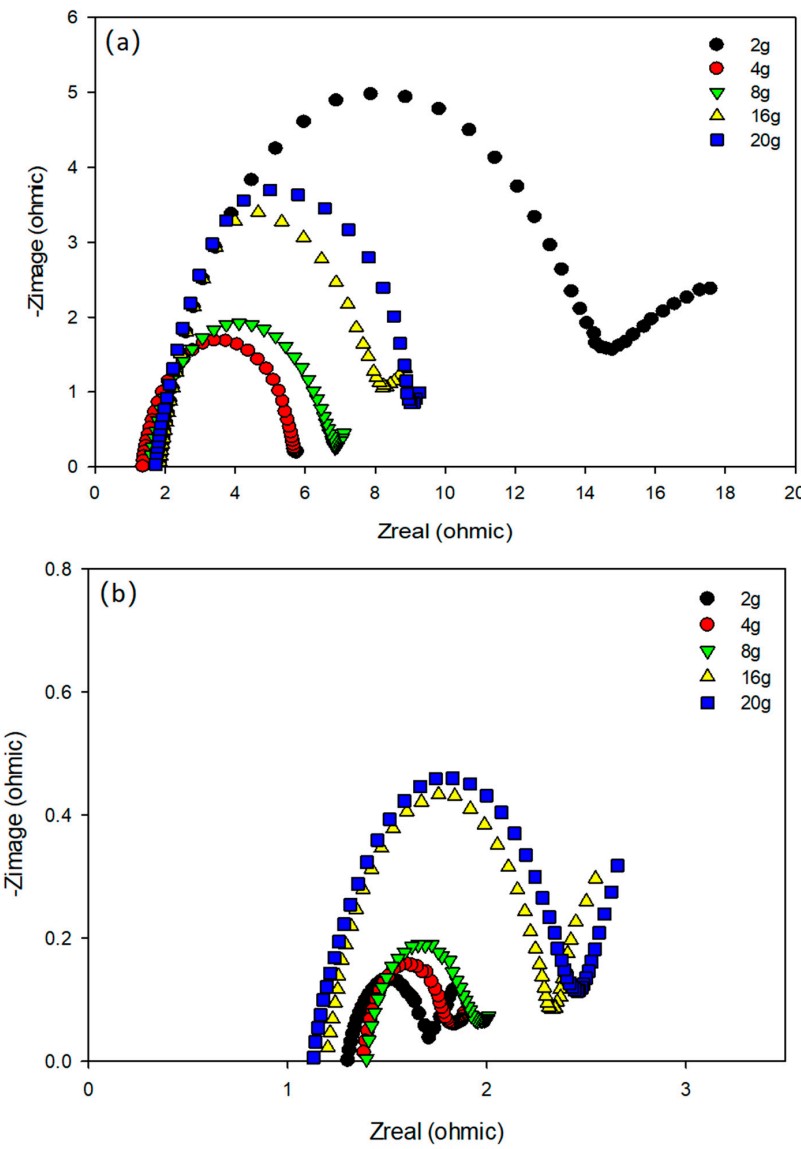

**Figure 6.** Nyquist plot of EIS data for (**a**) anode and (**b**) cathode in the BEAD reactor at different OLRs.

**Table 3.** Charge transfer resistance and Tafel slope for anode and cathode at different OLRs.

| | Anode | | | | | Cathode | | | | |
|---|---|---|---|---|---|---|---|---|---|---|
| ORL (COD/L.d) | 2 | 4 | 8 | 16 | 20 | 2 | 4 | 8 | 16 | 20 |
| Rs (ohm) | 1.37 | 1.52 | 1.36 | 1.85 | 1.73 | 1.42 | 1.45 | 1.31 | 1.19 | 1.13 |
| Rct (ohm) | 12.54 | 4.33 | 5.88 | 7.43 | 7.98 | 2.23 | 1.54 | 1.65 | 2.24 | 2.35 |
| C (uF) | 548 | 205 | 258 | 198 | 342 | 544 | 853 | 2631 | 2115 | 1524 |
| W (1/ohm sqrt(Hz)) | 5.67 | 1.44 | 0.985 | 8.01 | 4.85 | 4.54 | 8.76 | 9.36 | 20.16 | 15.32 |

### 3.5. Implications of Bioelectrochemistry for Brewery Wastewater Methanation

The nature and concentration of VFA is an important indicator for evaluating the status of anaerobic digesters. Generally, The effect of VFA on alkalinity in a conventional anaerobic digester was 0.3. However, in the BEAD reactor, when the OLR was increased from 0 g COD/L.d to 20 g COD/L.d, the concentration of VFA was gradually increased, but the VFA-to-alkalinity ratio was less than 0.17, which leading to process instability. OLR also contributes to increased COD and sulphate removal, with the highest COD removal of 96.5% occurring at an OLR of 2 g COD/L.d. The OLR of 2 g COD/L.d was used to

remove COD and sulphate. The highest sulphate removal of 91.3% was achieved at an OLR of 8 g COD/L.d. In this study, brewery wastewater was tested under OLR conditions with COD/L.d of 2, 4, 8, 16, and 20, respectively. Methane production increased slowly from 0.48 L/L.d to 5.64 L/L.d, which was significantly higher than that of the conventional anaerobic reactor as the OLR increased from 2 g/L.d to 20 g/L.d. The methane production of the brewery wastewater was significantly higher than that of the conventional anaerobic reactor. The performance of the conventional anaerobic reactor decreased significantly as the OLR increased to 16 g COD/L.d.

BEAD demonstrated efficient water treatment while greatly increasing the efficiency of anaerobic digestion for methane production. The removal of toxic hydrogen sulphide from the reaction environment can increase the rate and efficiency of methane production, although we did not further explore the changes in the concentration of hydrogen sulphide in the biogas. However, the BES was able to maintain good treatment efficiency even when treating complex brewery wastewater, which demonstrates the feasibility of BESs for intensive anaerobic digestion of brewery wastewater for methane production. There is no risk of anaerobic system breakdown, and cofermentation increases the buffering capacity of the system. BEAD significantly increased the rate and efficiency of $CH_4$ production. The effect of OLR intensifies the methanogenic process, and a modest increase in OLR can increase methane production under the same conditions. With the increasing organic load, the cumulative methane production is in an upward trend, which is consistent with the actual gas production pattern, and the methane production of BEAD is higher than that of the ordinary anaerobic system. The anaerobic digestion based on bioelectrochemistry can be easily put into practical applications and has high application prospects.

## 4. Conclusions

OLR is one of the most important process parameters reflecting the activity of methane production and the kinetic characteristics of the system. According to the results of the system gas production performance analysis as well as the system stability analysis, the organic loading has a positive effect on the anaerobic fermentation of brewery wastewater. The VFA concentration increased with the increase in COD concentration; when there was an increase in OLR from 8 g COD/L.d to 16 COD/L.d, the VFA concentration increased from 334 mg/L to 831 mg/L. When the OLR increased to 20 g COD/L.d, the ratio of VFA to alkalinity was still less than 0.17, which means that the system was more stable. Methane production also increased with increasing OLR, from 0.48 L/L.d at an OLR of 2 g COD/L.d to 5.64 L/L.d at an OLR of 20 g COD/L.d. The results indicated that the BEAD system could withstand higher OLRs and produce more methane compared to conventional anaerobic digestion. Therefore, the practical use of the BEAD system for biogas production from brewery wastewater is a promising option. It is important for the conversion of organic wastewater into renewable energy and allows for energy recovery and utilisation in industrial processes.

**Author Contributions:** Conceptualization, H.P.; methodology, Q.F.; validation, Y.Z.; formal analysis, X.L.; data curation, H.Z.; writing—original draft preparation, H.Z.; writing—review and editing, H.P.; visualization, H.P.; supervision, Q.F.; project administration, Q.F. All authors have read and agreed to the published version of the manuscript.

**Funding:** This research received no external funding.

**Data Availability Statement:** The data that support the findings of this study are available from the corresponding author upon reasonable request.

**Acknowledgments:** This research was supported by the National Natural Science Foundation of China [No. 21908118] and the Science, Education & Industry Integration Pilot Project [Grant No. 2022PY012], supported by Qilu University of Technology (Shandong Academy of Sciences), and the Technology Development Project [QL2023017] supported by Zhongding Special Metals QHD Technology Co., Ltd.

**Conflicts of Interest:** The authors declare no conflict of interest.

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
