# Peer review of "Influence of Organic Loading Rate on Methane Production from Brewery Wastewater in Bioelectrochemical Anaerobic Digestion"

_fermentation, doi:10.3390/fermentation9110932_

Round 1
Reviewer 1 Report
Comments and Suggestions for Authors
Dear Authors,
The manuscript ID: fermentation-2661777, entitled " Influence of organic loading rate on methane production from brewery wastewater in bioelectrochemical anaerobic digestion" aims to compare the effect of different organic loading rates (OLR) on the bioelectrochemical assisted anaerobic digestion of brewery wastewater in comparison with conventional anaerobic digestion. In general, the study is interesting, useful, and well-aligned within the scope of the Fermentation journal. However, in my opinion, significant revisions are needed before resubmission, particularly a comprehensive review of the English language (in particular in the Introduction and Results sections), as there are issues with poorly constructed sentences, sentence repetition, and loose information (in the wrong place).
In addition, other major revisions are necessary to improve the quality of the manuscript.
Keywords
The keywords should be revised so as not to repeat the words used in the title.
Introduction
Please define BOD5/COD in its first appearance and discuss the importance of this value.
Materials and methods
The units of TCOD should be included in the discussion. Please also include the information about the BOD.
Please, define CNT.
Since the aim is to treat wastewater, why did the authors aim to increase the hydrophobicity of the material surface?
How did the authors prepare the carbon material solution and why did they choose this concentration and applied voltage? Please refer to CNT instead of carbon material.
Please, indicate which material was used as the cathode and separator. Also indicate the voltage applied.
How did the authors prepare the different concentrations of the brewery wastewater? How much inoculum was used?
Please, include the conditions of the method used to measure the biogas.
Line 143, Please revise the temperature indicated.
How did the authors control the low pH of the wastewater? Please clarify this information throughout the manuscript.
Results and discussion
The colors used in Figure 3 are too similar, please change the colors to better distinguish between the acids.
Comments on the Quality of English LanguageIn my opinion, significant revisions are needed before resubmission, particularly a comprehensive review of the English language (in particular in the Introduction and Results sections), as there are issues with poorly constructed sentences, sentence repetition, and loose information (in the wrong place).
Author Response
Authors would like to thank the reviewer for the valuable comments. The manuscript was carefully revised point-to-point according to the comments, and all of the changes (highlighted) were included in the revised manuscript.

Reviewer 2 Report
Comments and Suggestions for Authors
Line 130 35±20°C
Questions for the future.
Didn't the Authors observe the formation of hydrogen at the cathode and oxygen at the anode? It is a pity that didn't track changes in the concentration of hydrogen sulfide in biogas. Didn't they observe the oxidation of hydrogen sulfide to elemental sulfur? Removing toxic hydrogen sulfide from the reaction environment increases the rate and efficiency of methane production. It might also be easier for them to explain the higher methane yield than Buswell's formula results from the reactions of hydrogen and carbon dioxide to methane.
Author Response

(The authors gave the same response as above.)

Round 2
Reviewer 1 Report
Comments and Suggestions for Authors
Dear Authors,
In my opinion, the manuscript has been sufficiently improved and is ready for publication at Fermentation journal.